# THE CERTIFICATION PARADOX: CERTIFICATIONS ADMIT BETTER EVASION ATTACKS

## ABSTRACT

In guaranteeing the absence of adversarial examples in bounded spaces, certification mechanisms play an important role in demonstrating neural network robustness. Within this work we ask if certifications themselves can potentially compromise the very models they help to protect? By demonstrating a new attack surface that exploits certified guarantees to construct norm minimising evasion attacks, we demonstrate the heretofore unexplored risks inherent in releasing certifications. Our new *Certification Aware Attack* produces smaller, more difficult to detect adversarial examples more than $74\%$ of the time than comparable attacks, while reducing the median perturbation norm by more than $10\%$. That this is achievable in significantly less computational time highlights an apparent paradox—that releasing certifications can reduce security.

## 1 INTRODUCTION

A well known property of learned models is that semantically indistinguishable samples can yield different model outputs (Biggio et al., 2013). When such samples are constructed deliberately these samples are known as *adversarial examples*, and they pose a significant risk to deployed models. This is especially true when the distance between clean and adversarial examples is minimised, as they can be incredibly difficult to detect. While *adversarial defences* have been proposed as a best response countermeasure to the adversarial attacks that generate these examples, the security they provide is often illusory, as motivated attackers can typically exploit-or-evade them.

In response to this dynamic, recent research effort has focused upon certified guarantees of adversarial robustness, which ensure that no adversarial examples exist within a calculable, bounded region (Weng et al., 2018; Zhang et al., 2018; Li et al., 2019; Salman et al., 2019b). While it is well-known that certified guarantees still admit practical attacks *outside* the region of certification, Table 1 demonstrates that only several works have considered how adversarial attacks may be applied to certified models, with the majority of these focusing upon how manipulations of the training corpus can be used to improve certifications at test time—a process that does not require reliable, hard to detect norm-minimising attacks. Cohen et al. (2019) stands alone as the are the only work to consider the risk of test-time attack against certified models, however they explicitly note that their tested attack does not align with the concept of attacking a certified model, and present no quantitative measures of performance. This paucity of test-time attacks, and the broad lack of consideration of what is required to attack a certified model leaves the security community without an understanding of how tight the guarantees provided by certification mechanisms are in practice, relative to the size of realisable adversarial attacks.

To delve into these risks, this defines how adversarial attacks can be applied to certified models, in a manner which allows extant attacks to be deployed against a range of certification mechanisms. Based upon this framework, we debut a novel attack surface that *certifications themselves can be exploited to construct smaller adversarial perturbations*. This attack, which we will henceforth refer to as a *Certification Aware Attack* exploits the very nature of certifications to help to rapidly identify adversarial examples. As Figure 1 demonstrates, the certifications *at identified adversarial examples* can then be exploited to guide the attack to new, norm-minimising adversarial perturbations. Thus the impact of our new attack framework is to *(i) speed up the initial stages of the search with larger and more informative jumps, and (ii) to reduce the total adversarial perturbation*, which ensures that these adversarial examples have a higher chance of avoiding detection (Gilmer et al., 2018).

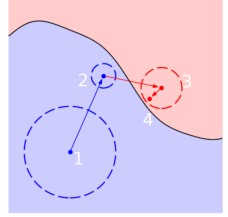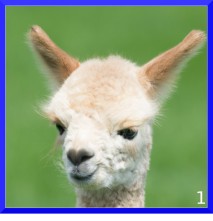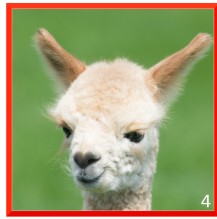

Figure 1: An illustrative evasion attack for a binary classifier, where knowledge of the certifications (circles) allows an iterative attack to better optimise their attack .

Table 1: Extant attacks, distinguished by if their goal is to change the label of samples or to just improve robust accuracy; if they were deployed at train- or test-time; if they have direct applicability to certifications (where half-circles denote attempting to improve certified robustness); and if they exploit the certifications themselves.

| Algorithms | Goal | Applicability | | | Exploits Certs. |
| | | Train | Test | Certi-fied | |
| --- | --- | --- | --- | --- | --- |
| PGD (Madry et al., 2018) | Label | ● | ● | ○ | ○ |
| Carlini & Wagner (2017) | Label | ● | ● | ○ | ○ |
| AutoAttack (Croce & Hein, 2020) | Label | ● | ● | ○ | ○ |
| DeepFool Moosavi-Dezfooli et al. (2016) | Label | ○ | ● | ○ | ○ |
| Training w/ Noise (Bishop, 1995) | Acc. | ● | ○ | ◐ | ○ |
| Salman et al. (2019a) | Acc. | ● | ○ | ◐ | ○ |
| MACER (Zhai et al., 2020) | Acc. | ● | ○ | ◐ | ○ |
| Cohen et al. (2019) | Label | ○ | ● | ● | ○ |
| Ours | Label | ◐ | ● | ● | ● |

## 2 BACKGROUND: CERTIFICATION MECHANISMS

Certification mechanisms eschew the responsive view of adversarial defences in favour of attempting to provide bounds upon the space in which adversarial examples $\mathbf{x}'$ can exist—typically a $\mathbf{x}$-centred $p$-norm ball of radius $r$ defined as $B_p(\mathbf{x}, r)$, where $r$ is strictly less than

$$r^\star = \inf \left\{ \|\mathbf{x} - \mathbf{x}'\|_p \colon \mathbf{x}' \in \mathcal{S}, F(\mathbf{x}) \neq F(\mathbf{x}'), \right\} \text{ where } F(\cdot) = \mathbb{1} \left( \arg\max_{i \in \mathcal{K}} f_i(\cdot) \right) . \quad (1)$$

where $\mathbb{1}$ is a one-hot encoding of the predicted class in $\mathcal{K} = \{1, \ldots, K\}$, and $\mathcal{S}$ is the permissible input space, which is typically $[0, 1]^d$ within a computer-vision context. The size of $B_p(\mathbf{x}, r)$ can be considered a reliable proxy for both the *detectability* of adversarial examples (Gilmer et al., 2018) and the *cost* to the attacker (Huang et al., 2011).

The construction of these bounds can be considered through either exact or high-probability methods. Exact methods typically construct their bounds by way of either Interval Bound Propagation (IBP), which propagates interval bounds through the model; or Convex Relaxation, which utilises linear relaxation to construct bounding output polytopes over input bounded perturbations Salman et al. (2019b); Mirman et al. (2018); Weng et al. (2018); Zhang et al. (2018); Singh et al. (2019); Mohapatra et al. (2020), in a manner that generally provides tighter bounds than IBP Lyu et al. (2021). However, these techniques exhibit a time and memory complexity that makes them infeasible for complex model architectures or high-dimensional data Wang et al. (2021); Chiang et al. (2020); Levine & Feizi (2022). While Lipschitz certified mechanisms Tsuzuku et al. (2018); Leino et al. (2021) have recently been proposed as a less computationally intensive alternative than bound propagation mechanisms, they still exhibit scalability issues for larger and more semantically complex models.

An alternate approach is the high-probability certifications of techniques employing *randomised smoothing* (Lecuyer et al., 2019), in which the Monte Carlo estimator of the expectation under

repeatedly perturbed sampling

$$\frac{1}{N}\sum_{j=1}^{N}F(\mathbf{X}_j) \quad\approx\quad \mathbb{E}_{\mathbf{X}}[F(\mathbf{X})] \quad\quad \forall i \in \mathcal{K} \tag{2}$$

$$\mathbf{X}_1,\ldots,\mathbf{X}_N,\mathbf{X} \quad\overset{i.i.d.}{\sim}\quad \mathbf{x} + \mathcal{N}(0,\sigma^2) \quad,$$

can be exploited to provide guarantees of invariance under *additive* perturbations. In forming this aggregated classification, the model is re-construed as a *smoothed classifier*, which in turn is certified. Mechanisms for constructing such certifications include differential privacy (Lecuyer et al., 2019; Dwork et al., 2006), Rényi divergence (Li et al., 2019), and parameterising worst-case behaviours (Cohen et al., 2019; Salman et al., 2019a; Cullen et al., 2022). The latter of these approaches has proved the most performant, and yields certifications of the form

$$r = \frac{\sigma}{2}\left(\Phi^{-1}\left(\breve{E}_0[\mathbf{x}]\right) - \Phi^{-1}\left(\widehat{E}_1[\mathbf{x}]\right)\right) \quad, \tag{3}$$

where $\Phi^{-1}$ is the inverse normal CDF, $(E_0, E_1) = \text{topk}\left(\{\mathbb{E}_{\mathbf{X}}[F(\mathbf{X})]\}, 2\right)$, and $(\breve{E}_0, \widehat{E}_1)$ are the lower and upper confidence bounds of these quantities to some confidence level $\alpha$ (Goodman, 1965).

## 3 ATTACKING CERTIFIED DEFENCES

While certified mechanisms provide guarantees of adversarial resistance, it is important to remember that these guarantees only exist to finite radii, and do not obviate the existence of all possible adversarial examples. This nuance may explain why there has not been any consideration in the literature of how certified models perform under attack.

When it comes to attacking certified models, one may intuitively think of simply identifying a sample $\mathbf{x}'$ such that $\|\mathbf{x}' - \mathbf{x}\| > r^\star$. However, in practice certification mechanisms are not able to construct tight bounds on $r^\star$, and even if they were, the search space for identifying $\mathbf{x}'$ would still be significant. Moreover, any such point would have an associated certified radii of $0$, which would likely trigger further inspection in any operationalised certification system. As such, within this work, we introduce the idea of a *confident* adversarial attack against a certification mechanism being one in which a certification constructed at the adversarial example is non-zero. In the case of a randomised smoothing based mechanism, such a condition is equivalent to

$$\arg\max \mathbb{E}_{\mathbf{X}}\left[F(\mathbf{x}')\right] \neq \arg\max \mathbb{E}_{\mathbf{X}}\left[F(\mathbf{x})\right] \quad\text{and} \tag{4}$$

$$\breve{E}_0\left[F(\mathbf{x}')\right] > \widehat{E}_1\left[F(\mathbf{x}')\right] \quad.$$

That this is highly concentrated (for sufficiently high Monte Carlo sample sizes) enables any of the reference attacks within Table 1 to be effectively employed *against the class expectations, rather than the individual draws under noise*. This contrasts with approaches like Expectation Over Transformation (Athalye et al., 2018), in which each sample under noise is attacked, in a process that requires significant numerically inefficient repetition of the attack process.

An additional complicating factor for randomised smoothing based mechanisms is that the final layer can be defined in terms of differentiable $\text{softmax}$ layers, or non-differentiable $\arg\max$ layers (Cullen et al., 2024). The latter of these is a core component of popular certification mechanisms, including that of Equation 3. It could naively be assumed that non-differentiable layers inherently defeat the gradient based attack mechanisms. However, this cannot be assumed to provide an insurmountable barrier to attack, with mechanisms including stochastic gradient estimation (Fu, 2006; Chen et al., 2019), surrogate modelling, and transfer attacks all providing potential avenues for a motivated attacker. In this work, we assume that the final $\arg\max$ layer can be replaced with a Gumbel Softmax (Jang et al., 2017)

$$y_i = \frac{\exp\left((\log(\pi_i) + g_i)/\tau\right)}{\sum_{j \in \mathcal{K}}\exp\left((\log(\pi_i) + g_i)/\tau\right)} \quad, \quad \forall i \in \mathcal{K} \quad. \tag{5}$$

Such a change has been chosen to maximise the difficulty of establishing gradients, while circumventing the need to assess if attack performance is a product of the chosen surrogate mechanisms for estimating gradients or the attack itself. However we must emphasise that the aforementioned attack definition is not reliant upon re-parameterising with the Gumbel-Softmax, and can be applied to models with a $\text{softmax}$ final layer, or indeed any other network architecture.

THREAT MODEL

For certified mechanisms that do not incorporate randomised smoothing—as seen in Section 5.3—our attack framework requires white-box access to the model, in a manner that requires oracle access to gradients, model predictions, certifications. Specific to attacks against models employing randomised smoothing, we also assume the attacker has the ability to construct derivatives through $\arg\max$ layers, and knowledge of the level of additive noise $\sigma$. However we note that even this last assumption is not strictly necessary, as Appendix F demonstrates that approximates values of $\sigma$ still provide enhanced certification performance.

# 4 CERTIFICATION AWARE ATTACKS

While there is value in understanding the tightness of certification mechanisms by attacking them with extant adversarial attacks, within this work we are also interested in understanding how certifications may be exploited by a motivated attacker to minimise the size of the identified examples. Such a concept may seem contradictory, but it is important to consider that from an attacker's perspective a certification can be viewed as a *lower bounds on the space where attacks may exist.*

In Section 4.1 we demonstrate how the existence of certifications at all points across the instance space (Cullen et al., 2022) can be exploited to significantly reduce the search space that must be explored to identify adversarial examples. Once an adversarial example is identified, Section 4.2 then demonstrates how certifications associated with successful adversarial examples can be *exploited to minimise the perturbation norm of the sample*, as any norm-minimising step inside the certified radii must still remain an adversarial example!

## 4.1 STEP SIZE CONTROL

We begin our attack process by solving the surrogate problem

$$\hat{\mathbf{x}} = \underset{\hat{\mathbf{x}} \in \mathcal{S}}{\arg\min} \left\{ |E_0(\hat{\mathbf{x}}) - E_1(\hat{\mathbf{x}})| \ : \ F(\hat{\mathbf{x}}) = F(\mathbf{x}) \right\} \ . \tag{6}$$

This formalism may seem counter-intuitive, as the constraint ensures that $\hat{\mathbf{x}}$ cannot be an adversarial example. However, consider the gradient-based solution of the previous problem

$$\mathbf{x}_{i+1} = P_{\mathcal{S}} \left( \mathbf{x}_i - \epsilon_i \left( \frac{\nabla_{\boldsymbol{x}_i} |E_0[\mathbf{x}_i] - E_1[\mathbf{x}_i]|}{\|\nabla_{\boldsymbol{x}_i} |E_0[\mathbf{x}_i] - E_1[\mathbf{x}_i]|\|} \right) \right) \ , \tag{7}$$

for a projection to the feasible space $P_{\mathcal{S}}$, and for which each $\mathbf{x}_i$ has associated certifications $r_i$. By imposing that $\epsilon_i > r_i$, we ensure that the new candidate solution $\mathbf{x}_{i+1}$ must exist outside the region of certification of the previous point, which is a *necessary but not sufficient* condition for identifying an adversarial example.

One approach for ensuring that $\epsilon_i > r_i$ would simply be to set the $\epsilon_i$ of Equation 7 to be

$$\epsilon_i = \rho(\mathbf{x}_i) \left( 1 + \delta \right) \ , \tag{8}$$

for some $\delta > 0$, and where $\rho(\mathbf{x}_i) = r_i$. However, doing so fails to account for the information gained from the certifications at all $\mathbf{x}_j$ for $j = 0, \ldots, i$. If we instead construct

$$\rho(\mathbf{x}_i) = \inf \left\{ \hat{\rho} \geq 0 : \mathbf{x}^\star(\hat{\rho}) \notin \bigcup_{j=0}^{i} B_P(\mathbf{x}_j, H\left[c_0 = c_j\right] r_j) \right\} \ , \tag{9}$$

$$\mathbf{x}^\star(\hat{\rho}) = P_{\mathcal{S}} \left( \mathbf{x}_i - \hat{\rho} \left( \frac{\nabla_{\boldsymbol{x}_i} |E_0[\mathbf{x}_i] - E_1[\mathbf{x}_i]|}{\|\nabla_{\boldsymbol{x}_i} |E_0[\mathbf{x}_i] - E_1[\mathbf{x}_i]|\|} \right) \right) \ ,$$

then the resultant step must remain strictly outside the certified radii region of all examples predicting the same class as the original sample point $\mathbf{x}_0$. Within this $c_i$ is the class prediction at step $i$ of the iterative process, and $H_{c_0=c_i}$ is an indicator function.

In practice taking such large steps may be disadvantageous in certain contexts, and as such in practice, we define $\epsilon_i$ in terms of pre-defined lower- and upper-bounds

$$\tilde{\epsilon}_i = \text{clip}\left(\epsilon_i, \epsilon_{\min}, \epsilon_{\max}\right) \ . \tag{10}$$

## 4.2 REFINING ADVERSARIAL EXAMPLES

Once we have identified an adversarial example, we switch to the second stage of our iterative process, in which we minimise the perturbation norm of any identified examples, in order to decrease their detectability. At this stage, the attack iterates $\mathbf{x}_i$ now produces a class prediction of $c_i \neq c_0$. Thus, any $\mathbf{x}_i$ must also be an adversarial attack if the difference between the two points is less than or equal to $r_i$. Thus our iterator can be defined as

$$\mathbf{x}_{i+1} = P\left(\mathbf{x}_i - \min\{\rho, \epsilon_{\max}\}(1 - \delta)\left(\frac{\mathbf{x}_0 - \mathbf{x}_i}{\|\mathbf{x}_0 - \mathbf{x}_i\|}\right)\right)$$

$$\rho = \sup\left\{\hat{\rho} \geq 0 : \mathbf{x}^\star(\hat{\rho}) \in \bigcup_{j=0}^{i} B_P(\mathbf{x}_j, H\left[c_0 \neq c_j\right] r_j)\right\} \quad (11)$$

$$\mathbf{x}^\star(\hat{r}) = P_{\mathcal{S}}\left(\mathbf{x}_i - \hat{\rho}\left(\frac{\mathbf{x}_0 - \mathbf{x}_i}{\|\mathbf{x}_0 - \mathbf{x}_i\|}\right)\right) \quad .$$

Similar to Section 4.1, a simpler variant of the above simply involves setting that $\rho = r_i$, however doing so discards the potential for prior certifications to help refine the search space. This framing ensures that $c_i = c_j \,\forall\, j > i$—ie. that all adversarial examples share the same class as the first identified adversarial example. As such, it may be true that there exists some adversarial example $\mathbf{x}''$

$$\|\mathbf{x}'' - \mathbf{x}_0\| < \|\mathbf{x}_i - \mathbf{x}_0\| \qquad \forall\, i \in \mathbb{N} \quad .$$

However, as we will now demonstrate, the process defined above still produces significantly smaller adversarial examples than other techniques, when each is employed against a certified model.

Algorithms detailing the aforementioned processes can be found within Appendix B, and the code associated with this work can be found at *Anonymised Link*. The details of the parameter space employed for our attack can be found within Appendix C.

## 5 RESULTS

To demonstrate the performance of our new Certification Aware Attack frame, we test our attack relative to a range of other comparable approaches. *Both our new attack and the reference attacks are deployed against models employing certifications, rather than the associated base classifiers.*

To achieve this, our experiments consider attacks against MNIST (LeCun et al., 1998) (GNU v3.0 license), CIFAR-10 (Krizhevsky et al., 2009) (MIT license), and the Large Scale Visual Recognition Challenge variant of ImageNet (Deng et al., 2009; Russakovsky et al., 2015) (which uses a custom, non-commercial license). In the case of models defended by randomised smoothing, each model was trained in PyTorch (Paszke et al., 2019) using a ResNet-18 architecture, with experiments considering two distinct levels of smoothing noise scale $\sigma$. Additional experiments involving against the MACER (Zhai et al., 2020) certification framework and a ResNet-110 architecture can be found in Appendix G. The confidence intervals of expectations in all experiments was set according to the $\alpha = 0.005$ significance level. To demonstrate the generality of our identified threat model, Section 5.3 eschews randomised smoothing to attack certifications constructed using IBP. Due to the inherent computational cost associated with constructing solutions with IBP, our results were limited to MNIST models solved using a sequential model of two convolutional layers followed by two linear layers, with ReLU activation functions. All calculations were constructed using NVIDIA A100 GPUs, with a single GPU being employed for experiments involving MNIST and CIFAR-10, while Imagenet test and training time evaluations were constructed using two GPUs.

## 5.1 ATTACKING RANDOMISED SMOOTHING

To establish the performance of our Certification Aware Attack framework against certifications employing randomised smoothing, Figure 2 explores the trade off between an attacks success rate, and the size of the identified adversarial examples for our approach relative to PGD and Carlini-Wagner. As each technique may be successfully attacking a different subset of samples, the size of the identified adversarial examples is normalised by their associated certified radii, as per Cohen et al..

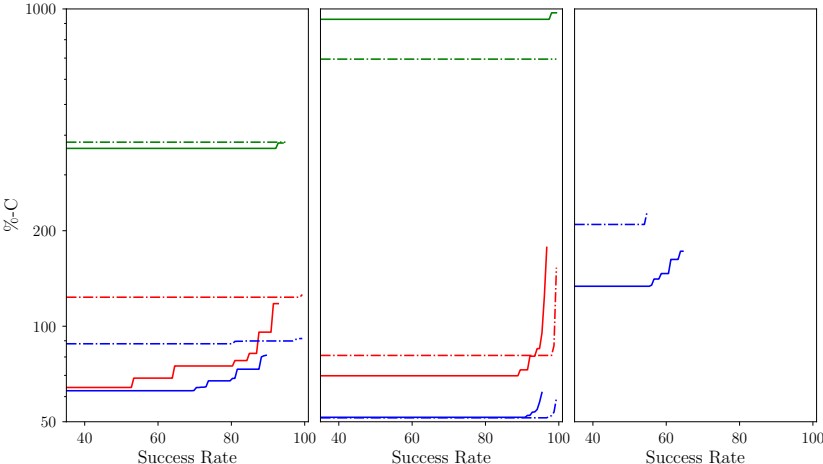

Figure 2: Minimum achievable average percentage difference between the attack radii and the certified guarantee of Cohen et al. (Equation 3) for a given success rate for our technique (Blue), PGD (Red), and Carlini-Wagner (Green), when tested against MNIST, CIFAR-10 and Imagenet. Solid and dashed lines represents $\sigma = \{0.5, 1.0\}$. Results calculated by sweeping across the parameters of Table 3.

The percentage difference between our approach and the certified radii, %-C, can then be considered as a measure of the average difficulty of identifying an adversarial example. These trade offs were achieved by considering performance over a broad sweep of each models respective parameter spaces, the details of which are included within Table 3 and Appendix C.

These results demonstrate a quasi-exponential relationship between the median percentage difference between the attack size to the location of the smallest possible *potential* adversarial example, which can be seen as a proxy for the ability of adversarial examples to be detected. Moreover, our approach consistently identifies significantly smaller adversarial examples than both PGD and Carlini-Wagner, with a 20 percentage point difference in the distance to Cohen et al. seen for CIFAR-10 at $\sigma = 1.0$, and an over 30 percentage point difference for MNIST. The one notable drawback of our approach is that over the parameter space tested we were unable to reach the same maximum success rates of PGD in the $\sigma = 0.5$ case, however we emphasise that PGD's distance to Cohen grows exponentially in the region where the success rate approaches $100\%$.

To consider a broader range of attacks, for the remainder of this work we will assume that an attacker employing employing either our Certification Aware Attack, PGD, or Carlini-Wagner would choose a point in parameter space that maintains a success rate above $90\%$, while minimising the median percentage difference to Cohen et al.. If such a success rate cannot be reached, it is instead assumed that the attacker chooses the point in parameter space which maximises the success rate.

To more comprehensively examine the suite of potential attack frameworks, we now expands our suite of comparisons to also include DeepFool (Moosavi-Dezfooli et al., 2016) and AutoAttack (Croce & Hein, 2020), which were excluded from broader parameter sweeps due to their relative performance. While AutoAttack has the ability to specify a $\ell_2$ norm perturbation magnitude, the associated computational cost makes a broader parameter space exploration infeasible. In contrast, while DeepFool is the fastest of all tested attacks, its failure to successfully identify norm minimising adversarial examples lead to its exclusion from a broader parameter exploration.

Across our full set of experiments, Figure 3 and Table 2 demonstrate that our new attack framework consistently constructs smaller adversarial perturbations than any other tested technique. On a sample-by-sample basis, in the most challenge experiment for our technique—Imagenet at $\sigma = 1.0$— our technique produces the smallest adversarial example for $54\%$ of the time (denoted by the *Best* column), for samples able to be attacked. This result is particularly striking given the relatively low success rate for our approach in Imagenet, relative to the other experiments, which suggests that the range of parameter space tested over may need further modification for datasets of the size and complexity of Imagenet. In the remainder of the tested experiments, as the success rate of our

Table 2: MNIST (M), CIFAR-10 (C), and ImageNet (I) attack performance across $\sigma$, covering the proportion of samples attacked (*Suc.*), smallest attack proportion (*Best*), median attack size ($r_{50}$), time (*Time* [s]), and percentage difference to the Cohen et al. (%-C)—additional details on these metrics can be found in Appendix D. All bar the success rate are only calculated over *successful attacks*. Tested attacks are Carlini-Wagner (C-W), AutoAttack (Auto) and DeepFool (DeepF). $\star$ denotes solutions selected following Appendix C, bolded values represent the best performing metric (excluding the success rate, as it is a control parameter), and arrows denote if a metric is more favourable with increases or decreases in the recorded value.

| | | $\sigma = 0.5$ | | | | | $\sigma = 1.0$ | | | | |
| | Type | Suc.↑ | Best ↑ | $r_{50}$ ↓ | %-C↓ | Time ↓ | Suc.↑ | Best ↑ | $r_{50}$ ↓ | %-C↓ | Time ↓ |
|---|---|---|---|---|---|---|---|---|---|---|---|
| M | Ours$\star$ | 90% | **73%** | **2.02** | 82 | 0.34 | 97% | **97%** | **2.23** | 90 | 1.22 |
| | PGD$\star$ | 91% | 19% | 2.17 | 96 | 2.04 | 99% | 3% | 2.62 | 123 | 2.03 |
| | C-W$\star$ | 93% | 7% | 5.46 | 364 | 3.03 | 95% | 0% | 5.36 | 380 | 3.02 |
| | Auto | 92% | 1% | 5.44 | 393 | 27.32 | 97% | 0% | 5.65 | 386 | 26.50 |
| | DeepF | 9% | 0% | 14.43 | 2417 | **0.07** | 51% | 0% | 17.10 | 2143 | **0.07** |
| C | Ours$\star$ | 91% | **87%** | **0.83** | 56 | 0.53 | 96% | **92%** | **1.26** | 56 | 0.86 |
| | PGD$\star$ | 92% | 4% | 0.92 | 72 | 2.17 | 99% | 3% | 1.46 | 77 | 2.15 |
| | C-W$\star$ | 98% | 5% | 3.13 | 432 | 3.18 | 99% | 1% | 3.65 | 352 | 3.14 |
| | Auto | 94% | 3% | 4.00 | 493 | 28.37 | 91% | 2% | 5.61 | 492 | 28.40 |
| | DeepF | 88% | 2% | 2.44 | 504 | **0.08** | 98% | 3% | 3.42 | 462 | **0.08** |
| I | Ours$\star$ | 65% | **74%** | **1.12** | 131 | 5.59 | 55% | **54%** | **1.30** | 160 | 5.69 |
| | PGD$\star$ | 81% | 19% | 2.03 | 214 | 52.82 | 92% | 39% | 3.17 | 193 | 52.07 |
| | C-W$\star$ | 52% | 4% | 32.93 | 5967 | 26.85 | 57% | 2% | 33.06 | 4240 | 28.43 |
| | DeepF | 56% | 3% | 2.45 | 661 | **3.01** | 70% | 5% | 4.41 | 646 | **3.04** |

technique increases, so too does the proportion of attacked samples for which our technique produces the smallest possible adversarial attack, demonstrating the viability of our approach as a framework for constructing minimal norm adversarial examples.

When considering the median certified radii produced by the techniques, our approach produces a median certification that is on average 11% smaller for MNIST, 12% for CIFAR-10, and 52% smaller in the case of Imagenet. Given that the approaches are not necessarily certifying the same samples, when controlling for the size of the certified radii of attacks, the %-C results are even more stark, which when compared to the next best technique in PGD exhibit an on average 24% reduction in the median size of identified adversarial attacks relative to their certified radii.

One feature noted within Section 4.2 was that all adversarial examples identified by our Certification Aware Attack framework must share the same class prediction as the first identified adversarial example. Intuitively it would appear that such a drawback would induce a disproportionate increase in the median certified radii as the dataset complexity is increased to the 1000-class ImageNet, as compared to MNIST or CIFAR-10. However in practice any disadvantages from this are outweighed by our attacks increased efficiency in exploring the search space. This efficiency in exploring the search space is evident in the computational cost of identifying attacks, with our approach requiring significantly less computational time to identify norm minimising adversarial examples relative to all of the other techniques, with the exclusion DeepFool. While DeepFool requires substantially less computational time than the other tested approaches, that it's percentage difference to Cohen et al. is orders of magnitude larger than other approaches emphasises that adding extra iterative steps to balance out the computational time would likely be a forlorn task.

Exemplars of some of these attacks can be seen in Appendix H.

## 5.2 TIGHTNESS OF CERTIFIED GUARANTEES

Performing these comparisons is useful not only for demonstrating the relative performance of each attack, but also for exploring the tightness of the bounds provided by the certification mechanisms. These bounds are by their nature *conservative* bounds against realisable adversarial attacks, and performing a comparison between these bounds and the actual minimal realisable adversarial attack can be considered evidence for the utility of certifications as a whole, and as a demonstration of

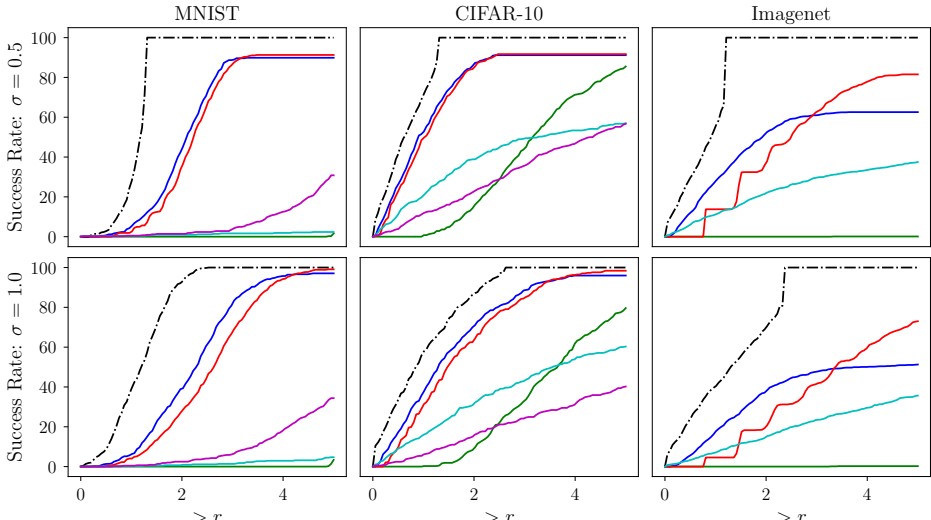

Figure 3: Best achieved Attack Proportion for our new Certification Aware Attack (blue), PGD (red), DeepFool (cyan), Carlini-Wagner (green), and AutoAttack (magenta); where the rows correspond to $\sigma = \{0.5, 1.0\}$ and the columns correspond to MNIST, CIFAR-10 and Imagenet. The black dotted line represents the theoretical best case performance following Equation 3.

potential avenues of future improvement for certification mechanisms. When considering these bounds in the contexts of the tested datasets, Figure 3 suggests that MNIST—which is often perceived as being the simplest of datasets—demonstrates a more significant delta between the certified radii and the attack performance. This may be a consequence of the simpler semantic properties of the dataset being more difficult to attack, relative to Cohen et al. style certifications.

The influence of $\sigma$ upon certification performance is especially interesting. Due to its role as a multiplicative constant in Equation 3, increasing $\sigma$ inherently increases the size of the certifications, an effect that is partially offset by a decrease in the observed class expectations. However, from the perspective of an attacker increasing $\sigma$ should also increase the smoothness of the gradients, which theoretically should make the model *easier to attack*. In practice, Figure 3 and Table 2 demonstrate that increasing $\sigma$ leads to a small increase in the size of identified attacks, relative to certified guarantees. While this may at first appear contradictory, it suggests that the ease in identifying adversarial attacks for larger $\sigma$ is offset by decreases in the tightness of the certified bound.

### 5.3 PERFORMANCE AGAINST OTHER CERTIFICATION MECHANISMS

To demonstrate the generality of our identified threat model, Figure 4 demonstrates the relative performance of our technique and PGD when tested for a model certified using IBP. While we have not fully explored the parameter spaces of both attacks, nor the broader suite of attacks in the context of this framework, these results reinforce the *information advantage* an attacker has when attempting to compromise models employing randomised smoothing *if they incorporate the certification into their attack*, irrespective of the certification mechanism. That this is true confirms that all certification mechanisms should assess their risk to adversarial attack in light of our Certification Aware Attacks.

## 6 DISCUSSION: BROADER IMPACT, LIMITATIONS, AND MITIGATIONS

While uncovering new attacks has the potential to compromise deployed systems, there is a prima facie argument that any security provided by ignoring new attack vectors is illusory. Such a perspective has uncovered new attack vectors including data poisoning, backdoor attacks, model stealing, and transfer attacks, all of which can now be protected against. Our work similarly reveals the paradox of certifications, in that the very mechanisms that we rely upon to protect models introduce new attack surfaces, enhancing the ability for attackers to construct norm minimising attacks.

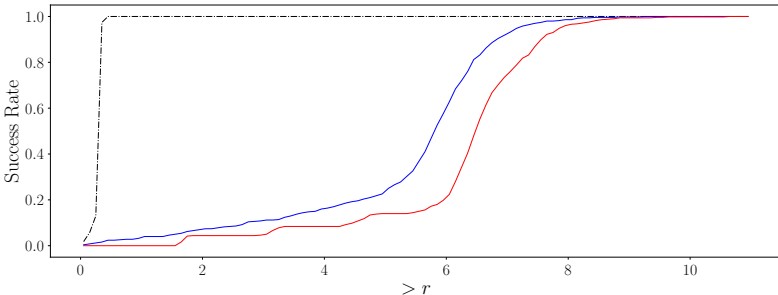

Figure 4: Success rate for our attack (blue) and PGD (red) for an IBP certified MNIST model.

It would appear that restricting the release of the certification could nullify our attack. However, the certifications can be trivially reconstructed from the class expectations and $\sigma$. Even if $\sigma$ is estimated, Figure 6b and Appendix F demonstrate that our attack still outperforms other frameworks. As such, we recommend that if possible, systems should only release the class predictions of certified models, without any associated expectations. While this would not prevent a suitably motivated attacker developing a surrogate model, it could increase the cost and difficulty associated with a successful attack. Exploring the effectiveness of this mitigation is left to future work.

It is important to while this work has specifically considered $\ell_2$ norm bounded evasion attacks against randomised smoothing and IBP based mechanisms, it is important to note that other threat models threat models (Liu et al., 2023), and certification frameworks exist, like Lipschitz certification Tsuzuku et al. (2018); Leino et al. (2021). However, these approaches still involve releasing a certificate, and this work has clearly demonstrated the existence of a novel attack vector against said releases. As such, we believe that any future certification—for evasion, backdoor, or other attacks—must consider the risk associated with the releasing certifications with significant care.

## 7 RELATED WORK

This work presents both a framework for attacking certifiably robust models, and a demonstration of how such certification can be exploited to improve attack efficacy. While we formalise the concept of attacking certifications, prior works have considered the impact of corrupting the inputs of both undefended and certified models. One common framework involves corrupting input samples with additive noise or adversarial examples, in order to improve robustness (Bishop, 1995; Salman et al., 2019a; Cohen et al., 2019). Of these, Salman et al. (2019a) is closest to our work, although their attacks only considered a small number of draws from randomised smoothing (rather than the full expectations), and employed a $\mathrm{softmax}$ in place of the $\arg\max$ operator. All three of these approaches are un-targeted, un-directed, training time modifications attempting to improve generalisation by increasing training loss. In contrast, our focus was placed upon both constructing a definition of test time adversarial attacks against certified models, and then exploiting the nature of certifications themselves to improve the performance of adversarial attacks against certified models.

## 8 CONCLUSION

We demonstrate the paradox of robustness certification: that tools for quantifying robustness (certifications) and defending base models (smoothing) can be exploited to support attacks against models. Through our novel Certification Aware Attack framework, we exploit this paradox to significantly decrease the size of the identified adversarial perturbations relative to state-of-the-art test-time attacks, leading to an up to $55\%$ decrease in the size of adversarial perturbations relative to the next best performing technique. Being able to reliably, repeatedly generate such norm-minimising adversarial examples would allow an attacker to reliably influence more samples before being detected than any other attack. These results underscore that significant consideration must be placed upon the safety of releasing robustness certificates.

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

## A BACKGROUND: ADVERSARIAL EXAMPLES

The existence of highly confident but incorrect adversarial examples in neural networks has been documented extensively (Szegedy et al., 2014; Goodfellow et al., 2015). We provide an overview of the topic in this appendix for completeness. Formally, adversarial examples are perturbations $\boldsymbol{\gamma} \in \mathcal{S}$ to the input $\mathbf{x} \in \mathcal{S}$ of a learned model $\mathbf{f}(\cdot)$, for which $F(\mathbf{x} + \boldsymbol{\gamma}) \neq F(\mathbf{x})$.

The $p$-norm of this perturbation can be considered a reliable proxy for both the *detectability* of adversarial examples (Gilmer et al., 2018) and the *cost* to the attacker (Huang et al., 2011).

The process for identifying such attacks commonly involves gradient descent over the input space. A prominent example is the Iterative Fast Gradient Sign Method (Madry et al., 2018; Dong et al., 2018), which we will henceforth refer to as PGD. This technique attempts to converge upon an adversarial example by way of the iterative scheme

$$\mathbf{x}_{k+1} = P_{\mathcal{S}} \left( \mathbf{x}_k - \epsilon \left( \frac{\nabla_{\mathbf{x}} J(\mathbf{x}, y)}{\|\nabla_{\mathbf{x}} J(\mathbf{x}, y)\|_2} \right) \right) \ . \tag{12}$$

This process exploits gradients of the loss $J(\mathbf{x}, y)$ relative to a target label $y$ to form each attack iteration, with the step size $\epsilon$ and a projection operator $P$ ensuring that $\mathbf{x}_{k+1}$ is restricted to $\mathcal{S}$.

---

**Algorithm 1** Certification Aware Attack Algorithm.

---

1: **Input:** data $\mathbf{x}$, level of additive noise $\sigma$, samples $N$, iterations $M$, true-label $i$, minimum and maximum step size $(\epsilon_{\min}, \epsilon_{\max})$, scaling factor $\delta \in [0, 1]$
2: $\mathbf{x}', \mathbf{x}'_s$ Successful $= \mathbf{x}, \mathbf{x}$, False
3: **for** 1 to $M$ **do**
4:     $\mathbf{y}, \breve{E}_0, \widehat{E}_1, r = \text{Model}(\mathbf{x}'; \sigma, N)$             ▷ Detailed in Algorithm 2
5:     **if** $\arg\max_{i \in \mathcal{K}} y = i$ **then**         ▷ Adversarial Example not yet identified.
6:         **if** $\breve{E}_0 > \widehat{E}_1$ **then**
7:             $\epsilon = \text{Equation 8 } (\mathbf{x}', \delta, \epsilon_{\min}, \epsilon_{\max})$
8:         **else**
9:             $\epsilon = \epsilon_{\min}$
10:         **end if**
11:         $\mathbf{x}' = \text{Equation 7 } (\mathbf{x}', \epsilon)$
12:     **else**
13:         **if** $r = 0$ **then**         ▷ Attempting to improve confidence of adversarial examples
14:             $\mathbf{x}' = P_{\mathcal{S}} \left( \mathbf{x}' + \epsilon_{\min} \frac{\nabla_{\mathbf{x}'}(\breve{E}_0 - \widehat{E}_1)}{\|\nabla_{\mathbf{x}'}(\breve{E}_0 - \widehat{E}_1)\|_2} \right)$
15:         **else**         ▷ Examples are refined while staying inside the certified radii
16:             $\mathbf{x}'_s$, Successful $= \mathbf{x}'$, True
17:             $\mathbf{x}' = \text{Equation 11}(\mathbf{x}', \delta, \epsilon_{\max})$
18:         **end if**
19:     **end if**
20: **end for**
21: **return** $\mathbf{x}'_s$, Successful

---

Carlini & Wagner (2017)—henceforth known as C-W—demonstrated the construction of adversarial perturbations by employing gradient descent to solve

$$\arg\min_{\mathbf{x}'} \left\{ \|\mathbf{x}' - \mathbf{x}\|_2^2 + c \cdot \max \left\{ \max\{f_\theta(\mathbf{x}')_j : j \neq i\} - f_\theta(\mathbf{x}')_i, -\kappa \right\} \right\} \ . \tag{13}$$

We note that while one-shot variants of these attacks have historically been used as a baseline for the performance of iterative attacks to be assessed against, we believe that by their nature such attacks always poorly represent the success-rate and attack-size trade off. Instead, we have performed our comparisons against the certified guarantee of Cohen et al. at the sample point, which provides an absolute lower bound on the size of possible adversarial attacks. We feel that this form of comparison more appropriately captures how these techniques perform, rather than attempting to compare one-shot with iterative attacks, which fundamentally incorporate different access level threat models.

## B   ALGORITHMS

Within Algorithm 1, lines 6–11 cover the processes outlined within Section 4 and 4.1, with lines 13–17 covering the materials of Section 4.2.

One important piece of detail relates to the case where $\breve{E}_0 < \widehat{E}_1$, which is equivalent to $r = 0$. Under both of these circumstances, the model is unable to construct a confident prediction, so the algorithm induces minimal size-steps either away from the origin—if an adversarial example has not yet been identified—or towards the most recent point, if that point was an adversarial example.

In order to calculate the class expectations and associated certifications for a given input $\mathbf{x}'$, Algorithm 2 performs the Monte-Carlo sampling and then corrects for sampling uncertainties. We note here that for the purposes of constructing derivatives, the lower and upper-bounding processes are treated as if they were perturbations to the expectations, and as such they are not considered as a part of the differentiation process. While this has the potential to slightly perturb the derivatives, our experiments have demonstrated that any $\delta > 0$ is sufficient to more than compensate

---

**Algorithm 2** Class prediction and certification for the Certification Aware Attack algorithm of Algorithm 1.

---

1: **Input:** Perturbed data $\mathbf{x}'$, samples $N$, level of added noise $\sigma$
2: $\mathbf{y} = \mathbf{0}$
3: **for** i = 1:N **do**
4: $\qquad \mathbf{y} = \mathbf{y} + GS\left(f_\theta\left(\mathbf{x}' + \mathcal{N}(0, \boldsymbol{\sigma}^2)\right)\right)$
5: **end for**
6: $\mathbf{y} = \frac{1}{N}\mathbf{y}$
7: $(z_0, z_1) = \text{topk}(\mathbf{y}, k = 2)$ $\qquad\qquad\qquad\qquad$ ▷ topk is used as it is differentiable, $z_0 > z_1$
8: $\left(\breve{E}_0, \widehat{E}_1\right) = (\text{lowerbound}(\mathbf{y}, z_0), \text{upperbound}(\mathbf{y}, z_1))$ $\qquad$ ▷ Calculated via Goodman (1965)
9: $R = \frac{\sigma}{2}\left(\Phi^{-1}(\breve{E}_0) - \Phi^{-1}(\widehat{E}_1)\right)$
10: **return** $\mathbf{y}, \breve{E}_0, \widehat{E}_1, R$

---

Table 3: Parameter space employed for our Certification Aware Attack, PGD (see Equation 12 for details), and Carlini-Wagner (see Equation 13).

| | | | |
|---|---|---|---|
| Ours | $\epsilon_{\min} \times 255$ | $=$ | $\{1, 5, 10\}$ |
| | $\epsilon_{\max} \times 255$ | $=$ | $\{20, 40, 100, 255\}$ |
| | $\delta$ | $=$ | $\{0.01, 0.025, 0.05, 0.075, 0.1\}$ |
| PGD | $\epsilon \times 255$ | $=$ | $\{1, 4, 8, 10, 20, 30, 40, 50, 100, 200\}$ |
| C-W | $c$ | $=$ | $\{10^{-5}, 10^{-4}, 10^{-2}, 10^{-1}, 1, 2, 3\}$ |

## C  PARAMETER SPACE

As was discussed in Section 5, understand the relative performance of techniques requires a consideration of how an attacks parameter space influences its performance metrics. In aide of this, for our three most highly performant attack frameworks, for each dataset we performed a parameter sweep over the parameters outlined within Figure 3. From this, for each attack we selected a representative position in parameter space that either exhibited the minimal %-C for a success rate over $90\%$, or, if such a success rate was not achievable, the maximum achievable success rate. In doing so, we attempted to construct fair comparisons that accurately reflected the performance of the techniques.

One complicating factor of such parameter sweeps is the computational cost associated with the exploration, especially in the case for Imagenet—as can be seen in Table 2. As such while we endeavoured to select our representative attacks based upon $500$ randomly selected samples, it was only possible to consider 50 samples for PGD and Carlini-Wagner for Imagenet due to the computational time associated with these parameter sweeps.

To explore the influence of the step-size control parameters of Equation 10, Figure 5 considers the influence of a range of these parameters upon key attack metrics, based upon the parameter space explored over Appendix C. Based upon this , it is clear that the primary driver of the success-rate and certification size trade off (as explored in Figure 2) is the parameter $\epsilon_{\max}$, that controls the largest possible step size that the Certification Aware Attack framework is allowed to make. Thus further exploring the parameter space in this direction would likely be a critical factor in increasing the success rate observed for Imagenet.

## D  METRICS

To help explore the relative performance of the tested techniques we consider a series of metrics which, in aggregate, reflect the overall performance of the technique. To explain these metrics in additional detail, the *Success Rate* represents the proportion of correctly predicted samples for which a technique is able to construct a successful attack, and can be calculated as

$$\text{Suc.}_i = \frac{1}{N}\sum_{j=1}^{N}(r_{i,j} > 0) \ . \tag{14}$$

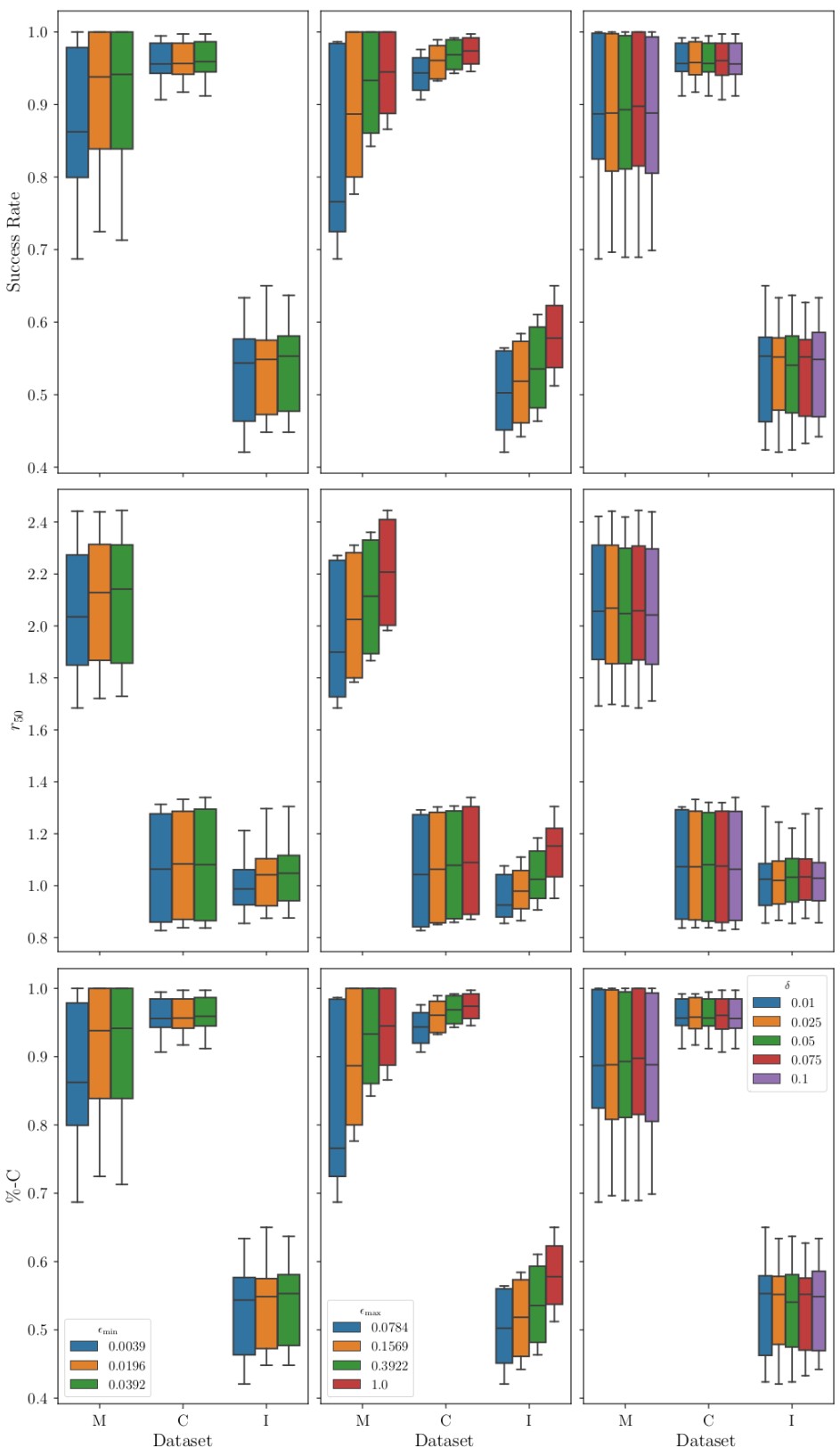

Figure 5: Response of key metrics for our Certification Aware Attack to changes in $\epsilon_{\min}$, $\epsilon_{\max}$ and $\delta$.

Here the subscript $i$ denotes a particular attack drawn from the set of attacks $\mathcal{I}$, and $r_{i,j} = \|\mathbf{x}'_j - \mathbf{x}_j\|$ is the attack radii, which for notational simplicity is set to $0$ in the case of a failed attack. The set of samples (of size $N$) has been filtered to ensure that each is correctly predicted by the model in the absence of an adversarial attack.

The *Best* is then the proportion of samples that a particular technique produces an attack radii smaller than any other correctly identified adversarial attack is calculable as

$$\text{Best}_i = \frac{\sum_{j=1} r_{i,j} \leq r_{i',j} \qquad \forall\, (i' \neq i) \in \mathcal{I}}{\sum_{j=1} r_{i,j} > 0 \qquad \forall\, i \in \mathcal{I}} \tag{15}$$

Increases to both of these metrics are advantageous, although as was noted in Appendix C each result within Table 2 must be contextualised against the decision to attempt to control the success rate to approximately $90\%$, if such an success rate was achievable for the technique in light of the tested parameter space.

The measure %-C represents the median percentage difference between the attack radii and the certified guarantee of Cohen et al., which takes the form

$$\text{\%-C} = \text{med}_{r_{i,j}>0}\left(\frac{r_{i,j} - C(\mathbf{x}_j)}{C(\mathbf{x}_j)}\right) \quad . \tag{16}$$

Here $\text{med}(\cdot)$ is the median over the set of successfully attacked samples, and $C(\mathbf{x}_j)$ is the certified radii for an $\ell_2$ norm, as calculable by Equation 3. Beyond this, $r_{50}$ is the median certified radii of the samples able to be successfully attacked by a given technique, and Time represents the median attack time (in seconds) across all tested samples. All three of these latter metrics demonstrate favourable performance with decreasing values.

This broad set of metrics was deliberately chosen to reflect different aspects of performance. However, we call particular attention to %-C, as it is a measure of the size of the adversarial examples *relative to the location of the minimal possible adversarial example*—with the certification of Cohen et al. essentially providing what is in essence characteristic scale that can be used for normalisation. We emphasise that such a measure of relative importance is important to further illuminate performance in light of the fact that the other metrics may not all strictly consider the same samples, as they are often constructed over the set of samples an attack method is successfully able to manipulate.

## E    SAMPLEWISE PERFORMANCE

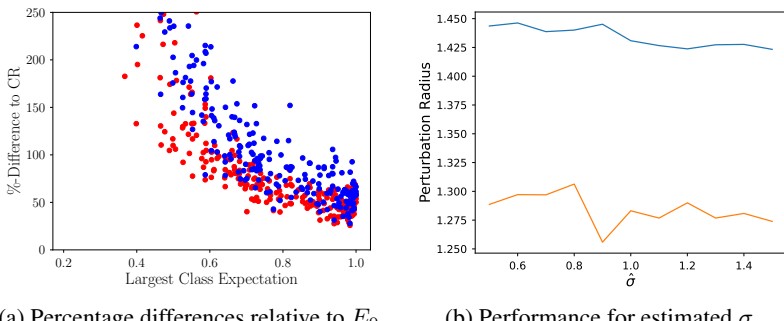

     (a) Percentage differences relative to $E_0$          (b) Performance for estimated $\sigma$

Figure 6: (a) captures the percentage difference between constructed adversarial perturbations and the certified radii of Equation 3 for CIFAR-10 at $\sigma = 0.5$, with Our technique in red and PGD in blue. (b) demonstrates that the blue mean and orange median performance of our technique are consistent even when $\sigma = 1.0$ is approximated by an estimated $\hat{\sigma}$.

To further illuminate the nature of the performance of our attack, Figure 6a considers the sample-wise performance of both PGD and our Certification Aware Attack. Within this data there is a clear self-similar trend, in which the percentage difference to Equation 3 increases as the largest class

expectation decreases. This difference could indicate the potential for improving the certification of samples within this region. There also appears to be a correlation between the outperformance of our approach and the semantic complexity of the prediction task, which suggests that tightening these guarantees could be increasingly relevant for complex datasets of academic and industry interest.

## F    ACCURACY OF $\sigma$

The white-box threat model assumes that the attacker has access to the full model and its parameters, including the level of additive noise $\sigma$. However, if the attacker only had access to the model and output class expectations, but was somehow prevented from directly accessing $\sigma$ and $r$, it turns out that the Certification Aware Attack can still be applied subject to a sufficiently accurate guess of $\sigma$. As is shown by Figure 6b, even over-estimating $\sigma$ by $50\%$ can decrease the radius of the identified adversarial perturbation under certain experimental conditions. That this is possible is a product of the terms $\delta_1$ and $\delta_2$ in Algorithm 1, as both of these parameters set the idealised step size to try and either change or preserve the predicted class. While this does suggest that there is potentially additional scope for optimising $\delta_1$ and $\delta_2$, it also demonstrates the possibility of estimating $\sigma$ as part of a surrogate model, in order to attack within a limited threat mode.

## G    TRAINING WITH MACER

Recent work has considered how certifications might be improved by augmenting the training objective to maximising the expectation gap between classes (Salman et al., 2019a). A popular approach for this is MACER (Zhai et al., 2020), in which the training loss is augmented to incorporate what the authors dub the $\epsilon$-robustness loss, which reflects proportion of training samples with robustness above a threshold level. In principle such a training-time modification can increase the average certified radius by $10$–$20\%$, however doing so does increase the overall training cost by more than an order of magnitude.

To test the performance of our new attack framework against models trained with MACER, Table 4 and Figure 7 recreate earlier results from within this work for CIFAR-10, subject to the same form of parameter exploration seen within Appendix C. We note that these calculations were performed with a ResNet-110 architecture, rather than the ResNet-18 architecture employed within the previous sections. While the broad qualitative feature of the success rates, best proportions, and median certifications broadly align with those seen within Table 2, we note that there is a significant difference in the %-C scores, which are a product of the ResNet-110 architecture (when trained under MACER) producing certifications that are an order of magnitude smaller than those observed within the main body of this work. That the attack radii are remaining constant while the certification radii decrease, strongly suggests that there would be significant scope for improving the performance of these results by varying the range of the parameter space exploration. One other notable feature is the improvement of DeepFool for MACER trained models, relative to the performance seen within the main body of this work, which we believe is a consequence of the changes in MACER's model decision space influencing the ability for DeepFool to converge upon successful evasion attacks.

## H    EXEMPLAR ATTACKS

The size of the associated adversarial perturbations has been established as a proxy of the risk of an adversarial attack evading human-or-machine-scrutiny Gilmer et al. (2018). While considering metrics of performance are a more reliable measure of this adversarial risk, for completeness in Figure 8 we have provided visualisations of the performance of both our attack and PGD. As both attacks share similar methodological features, the adversarial perturbations share similar semantic features, however our attack consistently requires smaller adversarial perturbations in order to trick the classifier—which in turn would have a higher probability of potentially evading any detection framework.

Table 4: CIFAR-10 attack performance across $\sigma$ for a ResNet-110 architecture trained with MACER. Table features follow Table 4

| $\sigma$ | Type | Suc.↑ | Best ↑ | $r_{50}$ ↓ | %-C↓ | Time ↓ |
|---|---|---|---|---|---|---|
| 0.25 | Ours$^\star$ | 100% | 76% | 0.83 | 2308 | 9.66 |
| | PGD$^\star$ | 100% | 5% | 1.03 | 2918 | 24.47 |
| | C-W$^\star$ | 24% | 0% | 9.10 | 39952 | 24.57 |
| | DeepF | 100% | 18% | 1.32 | 3687 | 7.01 |
| 0.5 | Ours$^\star$ | 77% | 58% | 1.09 | 2875 | 12.18 |
| | PGD$^\star$ | 95% | 18% | 1.73 | 2294 | 24.74 |
| | C-W$^\star$ | 43% | 1% | 11.35 | 19073 | 24.83 |
| | DeepF | 100% | 23% | 2.94 | 4377 | 7.58 |
| 1.0 | Ours$^\star$ | 59% | 43% | 1.38 | 12654 | 14.06 |
| | PGD$^\star$ | 98% | 39% | 2.86 | 3201 | 24.52 |
| | C-W$^\star$ | 9% | 0% | 9.63 | 20597 | 24.60 |
| | DeepF | 100% | 19% | 5.29 | 5670 | 7.11 |

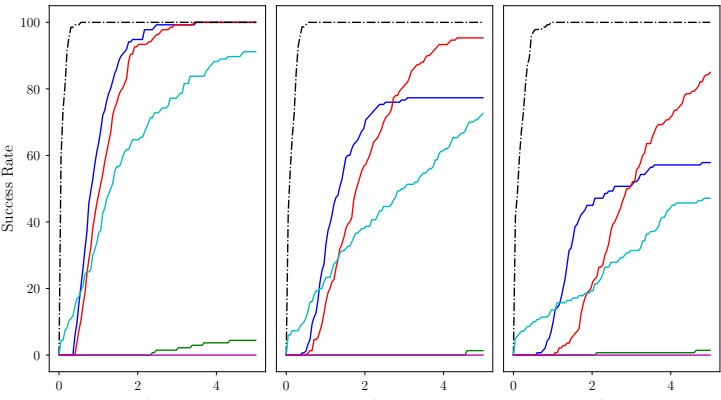

Figure 7: Attack and certification performance for a ResNet-110 model for CIFAR-10, when trained with MACER, covering our new Certification Aware Attack (blue), PGD (red), DeepFool (cyan), Carlini-Wagner (green), and AutoAttack (magenta). Similar to Figure 3, an ideal attack will approach the Cohen et al. (2019) radii suggested by the black dotted lines.

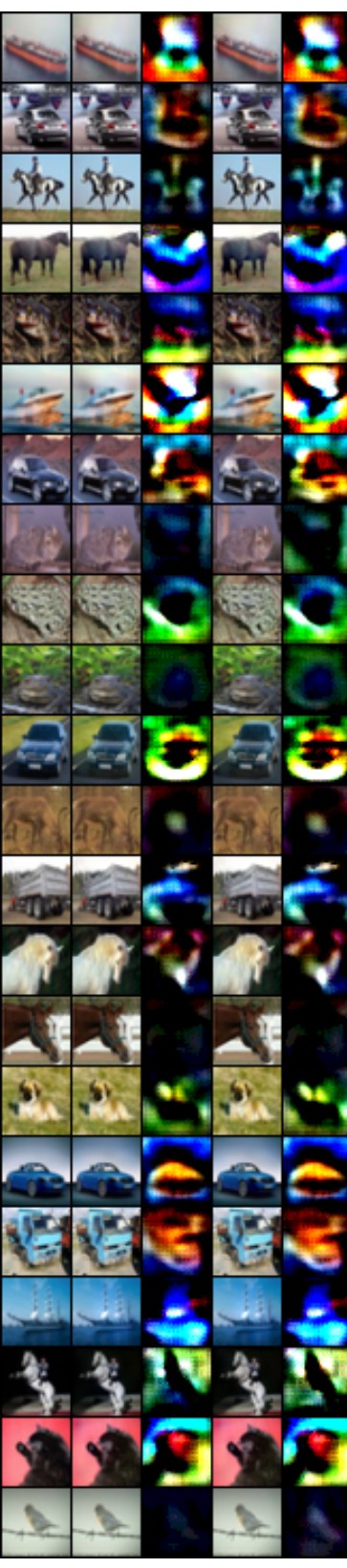

Figure 8: Illustrative examples of attack performance. Each column (from left to right) represents: the original image; the image under our attack; the adversarial perturbation associated with our attack; the image under PGD; the adversarial perturbation associated with PGD. The adversarial perturbations have been multiplied by 25 for visual clarity.

