# OpenReview forum: "The Certification Paradox: Certifications Admit Better Evasion Attacks"
_ICLR.cc/2024/Conference — Submitted to ICLR 2024_

### Official Review · Reviewer_xpHV · 2023-11-01

**Soundness:** 2 fair
**Presentation:** 2 fair
**Contribution:** 3 good
**Rating:** 3
**Confidence:** 3

**Summary:**

This paper proposes a technique for finding adversarial attacks given knowledge that a certification exists for a smaller radius.  The authors demonstrate experimentally on multiple datasets that compared to existing attack methods, their technique is able to consistently find attacks of minimum norm relative to the certification radius quickly.

**Strengths:**

- interesting, novel problem and proposed attack
- good experimental scope which encompasses multiple certified defenses, datasets, and attack techniques

**Weaknesses:**

- clarity: I think that the clarity of the experimental section can be improved greatly.  For figures, it would be useful to have a visible legend on the graphs as well as subtitles for each subplot so that they are more easily interpretable from a glance.  It would also help to section out and emphasize main observations since the text in the experimental sections gets quite long and it's easy to lose track of main conclusions.  In tables, it would help to also bold the best performing.  Some reported metrics are also a bit difficult to interpret, it might be useful to give some mathematical definitions (see questions)
- significance: From what I understand, the approach is finding attacks with radius larger than the certified radius and suggests that this is a security issue caused by certifications.  But this doesn't really come across to me as a problem with certifications as the paper seems to suggest that it is since the certifications give no guarantees for larger radii.  I still find the problem interesting though, but more from the perspective that we generally don't know exactly what perturbation size threshold is considered "imperceptible" and using a slightly larger perturbation size can still be an imperceptible attack that we want to defend against.  For this, it would be interesting to see visualizations for what the minimum norm adversarial example found by each attack technique looks like compared to the clean input.

**Questions:**

- to confirm, %C metric is the (distance to the attack found - certified radius) / certified radius ?
- what exactly is the "Best" metric in the tables?  I don't understand what the description "smallest attack proportion" means.
- how exactly are other attack methods used to find one of minimum norm?  For example, with PGD how do you decide on what radius ball to project to?

---

> ### Author Response · Authors · 2023-11-13
>
> We thank the reviewer for their time.
>
> We'd like to begin by addressing the point of significance: the existence of attacks outside the certified radius is a well understood property of certifications - certifications are, by their nature, conservative, and only have access to local information. Our contribution is to demonstrate that if the certification is released alongside the prediction, then this additional information can be exploited by an attacker to more efficiently construct adversarial examples, that are reliably smaller than attacks that don't consider this information. This is a property that has not been discussed in any certification literature to date, and represents a risk to the very kind of security conscious models that certification schemes attempt to defend.
>
> As to the commentary regarding that "a slightly larger perturbation size can still be an imperceptible attack" - yes, but the average attack size is a strong proxy to how detectable attacks are to *human or machine detection* - a property well established by Gilmer (and as cited within our paper). We believe that these metrics are more representative and fair for presenting how detectable adversarial examples against a model may be, relative to cherry picked visualisations of a small proportion of samples - especially when these visualisations would only be useful for considering how a human would view the perturbations, and not how automated systems might consider them.
>
> On the topic of clarity: all the details regarding the figures are discussed within the figure labels, and additional labelling on the figures themselves would require a commensurate decrease in the size of the figures to account for these details. We feel that the figure labels sufficiently detail these points. We would, however, be happy to bold the best-performing elements in Table 2.
>
> To address the questions raised in order:
> 1) Yes, the percentage difference between the attack and the corresponding certification guarantee of the original sample point is calculable as %-C = Mean( 100 * (||x' - x_0|| - C(x_0)) / C(x_0) ), where x' is the identified adversarial example, x_0 is the original sample, C(x_0) is the certification for the original sample, and || . || is the L2 norm (ie. the attack radii).
> 2) "Best" is the proportion of attacked samples for which an attack produces the smallest adversarial example. So Mean( r_i <= max(r_j for all j != i) ), where r_i represents the attack radius of a given technique.
> 3) As with our technique, the projection step is strictly limited to a projection into the feasible space for input samples.

---

> > ### Comment · Reviewer_xpHV · 2023-11-23
> >
> > Thank you for the response.  I appreciate the edits that were made in order to improve clarity.  I still believe that showing a few visualizations of the perturbed inputs using the proposed attack method would be beneficial (the perturbation should still visually preserve the identity of the input; if the smallest perturbation makes the input look like random noise or look like it belongs to a different class, why should we care about robustness against this attack?)
> >
> > Another reviewer also brought up the point that the attacks that are being compared against do not have minimizing the norm of the attack as the objective which might not be the best baseline of comparison.  I agree with this point.  If the authors could also incorporate attacks with the goal of finding the minimum radius attack, such as the one proposed in [1], I think that would greatly strengthen the comparisons in the paper.
> >
> > [1] Pintor, Maura, et al. "Fast minimum-norm adversarial attacks through adaptive norm constraints." Advances in Neural Information Processing Systems 34 (2021): 20052-20062.

---

> > > ### Author Response · Authors · 2023-11-23
> > >
> > > We would stress that an attack that minimises the norm of the adversarial perturbation will (on average) preserve the visual identity of the input more than samples constructed with an alternate attack. It's a well accepted principle within adversarial example research (see Gilmer's "Motivating the Rules of the Game for Adversarial Examples") that minimising the size of the average perturbation norm is equivalent to minimising the risk of human-or-machine based detection. That we are able to reduce the median perturbation norm by more than 10% over the next best performing attack means that we are, on average, producing samples that are more difficult to detect.
> > >
> > > While we maintain our earlier position that we believe any visualisation would be less instructive than our presented measures (given the potential for cherry-picking of results in the visualisations), we will endeavour to include additional visualisations in the final camera-ready submission. That said, we are not certain if the calculations will be completed before the time that the review period closes.
> > >
> > > As to norm-minising attacks, it is true that not all of the attacks we considered were originally formulated within a norm minimising framework. However, by constructing our metrics in terms of the smallest successful adversarial perturbation identified by each technique for a given sample (as we do), and then tuning our hyper-parameters to minimise the average perturbation norm of the attack across the test set, we do end up being able to minimise the average perturbation norm of the attacks. This is a well established process that has been used in previous attack papers - see for example Carlini & Wagner "Towards Evaluating the Robustness of Neural Networks", which takes this exact same approach.
> > >
> > > Given that we have resolved all of your issues relating to the clarity of our content and the associated metrics, we would ask if there would be any ground for you to potentially revise your score, especially given your appraisal that the work is interesting, novel, and that it contains robust experimentation.
> > >
> > > Thanks again

---

> > > ### Author Response · Authors · 2023-11-23
> > >
> > > Just as a brief follow up - we've now uploaded a rebuttal revision that includes exemplar images under attack using both our attack and PGD on pages 18 and 19.
> > >
> > > Thanks again for your consideration.

---

> ### Author Response · Authors · 2023-11-15
>
> We'd just like to take the opportunity to point out that we have submitted a rebuttal revision that we hope has addressed your issues and concerns. Specifically, we have revised the table caption for Table 2; have bolded the best-performing metrics within Table 2; have introduced Appendix D which explains each of the metrics used; and have updated Figure 3 to improve the accessibility of the figure.
>
> In light of your view of this work as being interesting, novel, and with sufficient experimentation, we would hope that our demonstrated commitment to resolving the issues raised within your review would elicit reconsideration of your rating. We would also welcome the opportunity to further discuss any issues that you may still hold.
>
> Thanks again.

---

### Official Review · Reviewer_Z9pA · 2023-11-01

**Soundness:** 2 fair
**Presentation:** 3 good
**Contribution:** 3 good
**Rating:** 5
**Confidence:** 3

**Summary:**

This paper investigates an iterative gradient-based method to obtain small-norm adversarial examples, where the perturbation norm $\eta$ (i.e., step size) at each iterate $x^t$ is determined dynamically by the certified radius around $x^t$ obtained using a robustness certificate leading to a speedup in iteration complexity. This speedup makes sense because the certificate ensures that no adversarial example having norm smaller than $\eta$ exists. Experiments demonstrate that this method finds smaller adversarial examples than baseline attacks.

**Strengths:**

1.	The idea of using certificates to speedup iterative gradient based attacks is interesting, and novel in my opinion.
2.	The reported experimental results improve a lot in percentage for finding minimal adversarial examples over existing approaches.

**Weaknesses:**

Most of these weaknesses stem from some surprising trends requiring further investigation in the main evaluation Table 2.

1.	Relevant Baselines 1: The subject of _minimal adversarial examples_ has been well studied in the literature, and some of these attacks are compared against in Table 2: C-W, AutoAttack. Other methods like PGD, DeepFool aim to find an adversarial example within the given budget, and not necessarily the adversarial example having minimum $\ell_p$ norm — it is unclear why these should be relevant baselines (see Sec 2.3 in [1] for more details). Now, AutoAttack is an ensemble of several attacks, out of which FAB is the attack created specifically for finding minimal adversarial examples, and hence this should be the primary comparison both for radius as well as time. There are a few parameters in FAB to tradeoff computational cost for norm-minimization (e.g., the number of random restarts), that should be ablated on.

2.	Relevant Baselines 2: If PGD is going to be used as a baseline, the hyper parameters tested in Appendix C are not enough: specifically, one should try a binary search on the perturbation norm that PGD is constrained to to find the minimum norm that PGD can find an adversarial example for (i.e., let Decision(s) be true if PGD can find an adversarial example in a s-sized ball around the input point. Then, one can binary search over Decision(s) to obtain the minimum s for which Decision(s) is true). This is potentially computationally expensive: see next point.

3.	Computational Overhead: Table 2 reports that PGD takes a larger time than the proposed method. It is very unclear how this can be true, since the proposed method can be seen as PGD + computing a robustness certificate at every step. Computation of certificates is typically extremely expensive (take orders of many seconds for large datasets), and as such it is quite unclear how can the combined time compare to any adversarial attack. A clearer description for exactly what the Time column corresponds to would be helpful in resolving this.

[1] Minimally Distorted Adversarial Examples with a Fast Adaptive Boundary Attack. Croce and Hein.

Additionally, some minor writing issues which don’t really affect the evaluation above, but are worthy of a mention:

1.	Use of the word “paradox”: Finding the minimal adversarial example is not paradoxical just because one is utilizing a certificate to compute the minimum norm. “Paradox” seems too strong of a word here.

2.	Eq. (9), typically I[..] is used for the indicator function. What is P in B_P? Since S is fixed throughout the paper, it might be easier notationally to just mention at the start that everything is projected onto S.

**Questions:**

Most of the questions stem from some trends in the main result Table 2 that are unclear (see details in weaknesses above). It would be good to clarify those questions.

---

> ### Author Response · Authors · 2023-11-13
>
> We thank the reviewer for their time and consideration.
>
> To address your points in order
> 1) With our baselines, we endeavoured to consider a representative set of attacks used within the literature. While there will always be more attacks that could be tested, we felt that this set served as an appropriate baseline for exploring the conceptual advantages that incorporating certification information within a PGD-style attack could bring. That FAB is included within AutoAttack left us confident that if AutoAttack was not producing a level of out performance that would justify a larger parameter exploration (given that it required an one-to-two orders of magnitude more computational time than other reference attacks), that a similar exploration with FAB would not yield beneficial results.
>
> We also would stress that our attack framework is a modification to PGD, but it is a modification that could similarly be applied to any other attack framework. As such, the information advantage gained by our attack approach should be most directly compared to PGD, and that the other tested attacks reinforce that our results are not an artefact of choosing to compare against PGD only.
>
> 2) All attacks - both ours, PGD, and C-W - could be embedded inside an optimisation regime that could improve their performance. However in doing so it would be incredibly difficult to assess the relative performance of the techniques in terms of their computational requirements. Moreover, doing so would remove our capacity to understand how choices in the parameter space influence the inherent trade-off between success rate and attack sizes (see the discussion around Figure 2).
> 3) As is discussed within the paper, both our attack and all the reference attacks are being deployed acting upon the certified model, not the base classifier. So the total time for each attack covers (# of attack steps before stopping) * ([Time to certify] + [Time to attack the certified model]). This is the case not just for our certification aware attack, but for all attacks, as we are specifically interested in the performance of adversarial attacks against models for which the certification is a core component.
>
> As to the minor points raised
> 1) The paradox of the title relates not to the ability of finding minimal adversarial examples, but rather that a mechanism associated with improving model robustness can release information that makes the very models it is seeking to defend easier to attack. This is a surprising and paradoxical result, and as such we feel the title is justified upon those grounds.
> 2) The P in B_P is the 2, as this is the only norm for which Cohen applies. This will be clarified in updates to the paper.
>
> With regards to table 2, as discussed above the time to attack is consistent, as all models are attacking the same certified model. To the other metrics, the success rates are a broadly controlled parameter (see Appendix C, set at ~90%); Best is the proportion of successfully attacked samples for which an attack produces a smaller attack than any other tested attack; the $r_{50}$ is the median attack radius; %-C is the mean percentage difference between the attack radius and the size of the certified guarantee at the original point - which is used in an attempt to normalise against the difficulty of constructing an adversarial example. For a controlled success rate, our technique reliably produces adversarial attacks that are consistently smaller than any of the other tested attacks, even when we consider this relative to the size of the certified guarantees. It does this while requiring less time than any other attack besides DeepFool - an attack that is barely ever able to produce a smaller adversarial attack than any other technique.

---

> ### Comment · Reviewer_Z9pA · 2023-11-13
> **Time Comparison**
>
> >As is discussed within the paper, both our attack and all the reference attacks are being deployed acting upon the certified model, not the base classifier. So the total time for each attack covers (# of attack steps before stopping) * ([Time to certify] + [Time to attack the certified model])
>
> Ok, then it is unclear to me how is this a fair comparison to PGD. Let us consider a (simplified) smoothed classifier $g_\sigma(x) = E_{v \sim N(0, \sigma)} f(x + v)$ produced by Randomized Smoothing. Say to evaluate $g_\sigma$, one uses $n_1$ samples from the gaussian, whereas to obtain a certificate for $g_\sigma$, one uses $n_2$ samples (in reality this is $C + n_2$ samples, because of statistical considerations see Cohen et. al. PREDICT vs CERTIFY). Since PGD's goal is to simply attack $g_\sigma$, one can get away with $n_1 \ll n_2$ - in other words, for simply finding an attack, one doesn't need to have a huge number of samples that is needed to actually compute the certificate (in fact $n_2$ is in practice huge for ImageNet-sized images). If $n_1$ is being set equal to $n_2$ as the authors suggest (which might approximately produce an equal time per inference step, even then it is unclear what happens to the additional $C$) then it seems that PGD is unnecessarily being penalized in time for computing a certificate at each step that is then thrown away. Using $n_1$ smaller than $n_2$ might show a tradeoff between time and adversarial attack strength for PGD, but I suspect the time falls much faster than the attack strength.
>
> To the above point of trade-off, see the discussion of $m_{\rm train}$ in Section 3, [A] where essentially the same issue is encountered: practically one can use a small $n_1$ to obtain adversarial examples against $g_\sigma$.
>
> [A]: Salman, Hadi, et al. "Provably robust deep learning via adversarially trained smoothed classifiers." Advances in Neural Information Processing Systems 32 (2019).

---

> > ### Author Response · Authors · 2023-11-13
> >
> > Sure, that's all reasonable **if the attacker can control the sample rate**, and is willing to accept the consequences of introducing this additional uncertainty to the sample. But we would ask what kind of certification deployment would allow the user to set the sample rate? If the attacker is able to set the sample rate, then that's a level of access that would also imply they could set things like $\sigma$ - and at which point, the certification framework essentially does not exist. We view the ability of an attacker to set the sample rate as being equivalent to being able to arbitrarily manipulate the models weights - which is well beyond the threat/access model considered here.
> >
> > Setting that $n_1 = n_2$ (following your introduced notation) allows for a fair apples-to-apples comparison with a reasonable level of access that is comparable to the other attacks.

---

> > > ### Comment · Reviewer_Z9pA · 2023-11-13
> > > **Baselines relevant to minimal perturbation norm**
> > >
> > > Agreed, but _level of access_ is a deeper discussion that is out of scope here (e.g., is even access to gradients of the network a reasonable level of access, let alone the actual certified radius at each point? ). I respectfully disagree with the authors' stance of not including such comparisons for this reason, and would leave to other reviewers to decide.
> > >
> > > Regarding the other aspect of baseline comparisons,
> > > > All attacks - both ours, PGD, and C-W - could be embedded inside an optimisation regime that could improve their performance. However in doing so it would be incredibly difficult to assess the relative performance of the techniques in terms of their computational requirements.
> > >
> > > It is unclear whether the performance gains would be comparable, as the proposed method is created precisely to obtain a minimal perturbation norm, but PGD is not (as other reviewers mention too). In fact, I think that having a simple optimization procedure on top of baselines, to obtain the minimal perturbation norm, would reduce the search space for the baselines, making the performance comparison clearer.

---

> > > > ### Author Response · Authors · 2023-11-13
> > > >
> > > > We would strongly contend that access to the gradients is a standard white-box attack assumption that is well aligned with the established norms of the adversarial attack community. We view the assumption that the certifications (or even the class expectations, which can then be used to reconstruct the certifications) are released to not be unreasonable, because until our work no paper has posited any security risks associated with their release.
> > > >
> > > > Our work is aligned with community expectations regarding what is considered a reasonable level of access to assume - both due to the risks of gradients being directly released, or being reconstructed through transfer attacks. However we would strongly contend that the idea that the attacker has the ability to set the sample count to be a proposition that is well beyond the standard threat models.
> > > >
> > > > Moreover, our technique could also involve a decrease in the sample count - there is no part of this process that relies upon the sample count being large, and reducing the sample count would still give an information advantage to our attack algorithm, an information advantage that as we show in this work allows the attacker to construct better adversarial examples. However, we set the sample count at a level that reflected what we considered to be an indicative amount for any deployed system that would employ randomised smoothing, based upon our view of the literature.
> > > >
> > > > As to your point regarding optimisation: it may be unclear if the performance gains would be comparable, but the increases in computational cost would be significant. Take AutoAttack for example - if an optimisation routine requires 20 queries to set its parameters for a given sample (the same amount as was used in Carlini & Wagner), then you're talking 10 minutes per attacked sample, or 83 hours to attack 500 samples with AutoAttack, and 332 hours across MNIST and Cifar-10 (ignoring imagenet) datasets and choices of $\sigma$. Carlini-Wagner would involve 183 hours. PGD 314 hours. So just under 5 weeks of computational time for those 3 attacks, and that's before we add our attack to the mix as well. At what point does this become an unreasonable proposition? Even if it was 10 queries per sample for the optimisation routine, across those 3 attacks parameter optimisation would require 414 hours The additional computational cost that comes from attacking a model employing randomised smoothing inherently limits us from taking such an exploration
> > > >
> > > > Our choice to consider parameter sweeps rather than optimisation was a deliberate one, as it allowed us to considered the relative performance of 3 of the key attacks (with AutoAttack excluded due to its computational time), and gave us the ability to construct fair comparisons in a manner that respected a computational budget. An additional advantage of such a sweep is that it allowed us to explore the trade off between attack size and success rates, whereas optimisation alone would only consider attack sizes.

---

### Official Review · Reviewer_VDGn · 2023-11-02

**Soundness:** 3 good
**Presentation:** 2 fair
**Contribution:** 3 good
**Rating:** 6
**Confidence:** 2

**Summary:**

The paper presents a thought-provoking analysis of the potential vulnerabilities introduced by the release of certification mechanisms in neural networks. It posits an important question about whether certifications, while intended to establish the absence of adversarial examples within bounded spaces, might paradoxically undermine the security they are designed to bolster.

Through experimentation, the authors introduce the novel "Certification Aware Attack," which adeptly exploits the provided certifications to mount evasion attacks that minimize the perturbation norm. This approach is shown to generate adversarial examples that are notably smaller and more challenging to detect than those produced by existing attack methods. The paper claims that such attacks can be executed with a reduction in the median perturbation norm, implying that adversaries could create more discreet yet effective perturbations. The authors further claim that these sophisticated attacks require less computational time, highlighting a security trade-off when releasing certifications.

The paper could impact our understanding of neural network security, suggesting a re-examination of the strategies employed to certify model robustness. It underscores the complexity of defending against adversarial attacks and the unintended consequences that well-meaning security measures can provoke. The findings could catalyze a reassessment of current practices and prompt the development of more resilient security protocols in machine learning models.

**Strengths:**

1. The idea of this paper is interesting, and it explores a new setting in the certified adversarial machine learning area.
2. The methodology introduced in this paper is valid to me, and the formulation is clear
3. The authors conducted sufficient evaluations to demonstrate that the required distance can be significantly reduced by their proposed attack method.

**Weaknesses:**

1. Although it is an interesting setting to me, the practicality of this paper is questionable.
2. The presentation of the paper could be improved. Some symbols are not defined or hard to understand.

**Questions:**

Please see the weakness section

---

> ### Author Response · Authors · 2023-11-13
>
> We thank the reviewer for their time and consideration, and appreciate that they found our work interesting, valid, and clear.
>
> We would, however, disagree with the contention that our paper is impractical - it requires no additional steps beyond what those would be required to apply any adversarial attack against a model employing randomised smoothing (or indeed, any other certification mechanism), and in doing so we are able to demonstrate that any certified model that releases its certifications is inherently providing an attacker with information that can be exploited to better help them craft minimal norm adversarial examples. That this is possible is important, as the size of constructed certifications is often significantly smaller than the smallest adversarial example that exists (see Figure 3), and these samples are often small enough to evade both human and machine detection.
>
> We would, however, be happy to add additional detail regarding symbols. From the other reviews it would appear that our metrics "%-C" and Best would be enhanced by providing the mathematical explanations (rather than the current written explanations), and we will add these to the paper.
>
> Thanks again.

---

> ### Author Response · Authors · 2023-11-15
>
> We'd just like to call attention to our recently uploaded rebuttal revision, which among other changes has introduced Appendix D, which we have used to help define some of the symbols that other reviewers have raised as being potentially ambiguous.
>
> Thanks again for your consideration.

---

> > ### Comment · Reviewer_VDGn · 2023-11-20
> > **Response**
> >
> > Thanks for the rebuttal. After reading it and other reviews, I decided to keep my rating as 6.

---

### Official Review · Reviewer_Kwhk · 2023-11-07

**Soundness:** 2 fair
**Presentation:** 2 fair
**Contribution:** 3 good
**Rating:** 1
**Confidence:** 5

**Summary:**

This paper proposed a stronger adversarial attack on neural network classification models. The core idea is to use a certification method (randomized smoothing) to estimate a lower bound of radius where adversarial examples cannot exist, and then focus on searching for adversarial examples only in the regions where certification fails. I do believe this idea, if demonstrated correctly, can be helpful for finding adversarial examples, because with the help of strong certification algorithms, the region that does not include adversarial examples can be excluded and the search space is reduced.

While this approach is novel and interesting, the main storyline in this paper is rather misleading - it claims that certified models are more prone to attacks, and claims that certification reduces security. These claims are too extreme and biased.

Despite presenting the results as a big surprise, the result presented in this paper is not really a surprise because certification and attacks essentially solve the same problem, and the strongest certification method also leads to the strongest attack method. For example, using a mixed integer programming solving can give (theoretically) the strongest certification, as well as giving the strongest attack, as it leads to the global optimal solution to the adversarial attack problem.

In other words, if we have a strong certification algorithm that can give very tight bounds, any part of the input space where the bounds do not hold will necessarily contain adversarial examples by definition. In the author’s claim, a stronger certification algorithm like this will reduce security, which apparently does not make sense to me; instead, a stronger certification algorithm helps us to easily find which part of the model is robust and which part is not, and with this exact quantification of the model, we have more security rather than less. The real security problem is that we don’t know whether the model is secure or not, because in many cases we can neither certify nor attack them. This is also the actual problem certified models try to resolve - at least, we have guarantees in some cases to know the model is indeed safe. This paper does not reduce the provable security guarantees provided by certified models, despite its misleading claims.

I like the novel approach to the adversarial attack problem, but the paper cannot be accepted in its current form because of its extreme, misleading, and confusing claims. None of the theoretical security guarantees in certified defense were actually broken by this paper, and the usage of the certification method in adversarial attacks shows the power and tightness of certification itself. If accepted, it will create misunderstanding in the research community. I am happy to reevaluate the paper if the authors are willing to significantly rephrase the paper and rewrite the story to tune down the misleading overclaims.

-----------------------

After discussions with the authors, despite multiple reviewers raising the same concern as me, the authors did not want to change their misleading and confusing claims. Accepting this paper in its current form could lead to confusion and even damage to the certification community. Thus, I have to further reduce my score and firmly reject this paper.

**Strengths:**

1. The method of using a certification algorithm to guide the procedure of finding adversarial examples is novel and not well explored in prior works. If the paper can be written in a different way, by emphasizing that certification tools are also useful for guiding adversarial attacks and giving better quantification of model characteristics from both attack and certification perspectives, it can become a good paper.

2. The topic studied in this paper is important and relevant. The proposed method also demonstrates improvements in some metrics (mostly the perturbation sizes, but not attack success rates) on some models.

**Weaknesses:**

1. The claims in this paper are misleading and confusing (see my comments above).

2. Many samples are used to estimate $E_0$ and $E_1$ during the attack. To make a fair comparison, for other attacks such as PGD, AutoAttack and CW, the number of random restarts must be adjusted, so each baseline contains the same numbers of queries to the model. With more random restarts, other attacks also become stronger.

3. Attacks like AutoAttack are designed for a fixed norm, so I don’t think the comparison on minimal perturbation size (which is one main claim of the proposed method) is fair here. Attacks like CW have hyperparameters to tradeoff between attack success rate and the norm of the adversarial examples. The table just reports one number without showing the tradeoff or selection of the norm for baseline methods, so it is not convincing that the proposed approach is stronger.

4. PGD was usually a strong baseline for L inifty norm. Since this paper focus on L2 norm attack, the improved version of PGD attack for L2 norm should be used as a baseline [1].

[1] Rony, Jérôme, et al. "Decoupling direction and norm for efficient gradient-based l2 adversarial attacks and defenses." Proceedings of the IEEE/CVF Conference on Computer Vision and Pattern Recognition. 2019.

**Questions:**

1. Why does the proposed approach achieve much worse performance in MACER models in the appendix? (attack success rate is much lower than PGD). How about other certified defense methods such as SmoothAdv?

2. Section 5.3 briefly mentioned IBP models. Did you also use the IBP bounds to guide attacks? For example, you can search for the region where IBP cannot certify.

3. Is the “%-C” metric based on the certified radii of the adversarial examples being wrong classified? Can you give a clear mathematical definition of this metric?

4. In Figure 3, why do the imagenet results with PGD have weird steps, while other baselines do not?

---

> ### Author Response · Authors · 2023-11-13
>
> We thank the reviewer for their consideration.
>
> We'd like to begin by addressing the claims the reviewer in the fourth paragraph of their summary. We emphasise that our paper does not produce a certification algorithm. As stated in the abstract, we explore "the heretofore unexplored risks inherent in releasing certifications" by exploiting the fact that the release of a certification gives an attack additional information that they can exploit to find the adversarial examples that must exist outside the regions of certification. The risks associated with stronger certification mechanisms is that they provide more relevant information about where adversarial examples cannot be, which in turn would increase the risk to the certified model being exploited following our algorithm.
>
> We are confused as to statements like "this paper does not reduce the provable security guarantees provided by certified models, despite its misleading claims" - we make no claim of breaking the provable guarantees of certified models. These certifications provide tight guarantees on where adversarial examples cannot exist, but they do not provide a tight bounding on where adversarial examples do exist - and this difference is significant (Figure 3). Certified defences are still vulnerable to attack, and the size of certificates of robustness are often not large enough to eliminate the potential of difficult to detect adversarial examples.
>
> We are also curious as to which claims the reviewer finds to be misleading? This is a strong statement that we would happily rebut. Our paper consistently and clearly states that releasing certifications allows for an attacker to exploit the certifications to identify *minimal norm adversarial attacks outside the radius of certification*. None of this point relates to breaking the guarantees of certified robustness.
>
> To address the weakness relating to success rates - Figure 2 demonstrates that there is an inherent tradeoff between success rates and the size of the identified adversarial examples, and that our technique outperforms all other techniques when considereding this tradeoff (see Figure 2). The Table 2 data that the reviewer is referring to on this point must be considered in context of Figure 2 and Appendix C, where we discuss that the points in parameter space for Table 2 were chosen so that each technique produced a success rate of 90% if possible. This also relates to the third identified weakness. While parameter sweeps like those used for our attack, C-W, and PGD would also be possible for AutoAttack, we emphasise that AutoAttack required  an order of magnitude more time than any other attack for MNIST and C-W - so performing this kind of parameter exploration would have been prohibitively expensive and unrealistic for any attacker.
>
> As to the second weakness, that "many samples are used to estimate E_0 and E_1 during the attack" - we reiterate the point made within the paper that each attack (be it ours, PGW, C-W, etc.) is applied to the smoothed variants of the models outputs - so after E_0 and E_1 have been estimated. Each reference attack is applied to the exact same output, the only difference is that our attack also takes into account the size of the robustness certificate at the same time.
>
> Finally - we do use the L2 variant of PGD.
>
> To cover the questions in order:
> 1) Against MACER our technique exhibits almost identical step median radii, relative computational time, and the a similar ratio between the amount of samples it is able to certify and the amount of samples for which it is the best technique. The percentage distance to Cohen changes for all techniques because MACER is certifying a greater proportion of samples, but oftentimes those additional samples are so small that they distort the %-C metric for all attacks. With regards to the proportion able to be attacked - we believe that the settings from Table 3 may not be sufficient to account for the change in the distribution of certifications that MACER produces.
> 2) Yes, the exact same attack framework is applied - the region guaranteed to contain no adversarial examples is used to guide the step size for our certification aware attack.
> 3) No. %-C is the percentage difference to Cohen et. al.: %-C = Mean( 100 * (r(x', x0) - C(x0)) / C(x0) ), where r(x',x_0) is the distance between the adversarial and original samples and C(x0) is the certification for the original sample.
> 4) As no technique was able to reach the 90% success rate for Imagenet, so the parameter that maximised the success rate was chosen. This appeared to produce this interesting clustering for the samples that admitted small attacks, which we believe that this is a consequence of how the loss landscape behaves for randomised smoothed models - noting that this same behaviour is not visible for IBP, or any other dataset.

---

> > ### Comment · Reviewer_Kwhk · 2023-11-23
> > **Thanks for the response**
> >
> > Thank you for the response and discussion. I appreciate your clarifications on some technical details.
> >
> > My main concern is also echoed by Reviewer xpHV:
> >
> > >  From what I understand, the approach is finding attacks with radius larger than the certified radius and suggests that this is a security issue caused by certifications. But this doesn't really come across to me as a problem with certifications as the paper seems to suggest that it is since the certifications give no guarantees for larger radii.
> >
> > The authors asked which part of the paper was misleading or confusing. The title is already confusing enough, and the entire story is presented in the way that "certifications can reduce security". In fact, most people would agree that certification gives us security and assurance of a system. Without certification, we have security by obscurity or no security at all. I read the latest version of the paper and none of these confusion claims were changed.
> >
> > Also, my concerns on several technical questions were not addressed, such as the unfair comparisons of the size of the adversarial perturbations to methods that do not aim to minimize the perturbation size. No results were shown for the L2-specific PGD attack "Decoupling direction and norm for efficient gradient-based l2 adversarial attacks and defenses."
> >
> > In conclusion, I believe the current version of this paper is problematic and not publishable. I hope the authors can consider comments from me and other reviewers and rewrite the story.

---

> > > ### Author Response · Authors · 2023-11-23
> > >
> > > Thank you for your comment. We don't disagree that certifications have the potential to play an important role in securing machine learning. However, as emphasised in sections 6 and 8 **releasing** these certifications has the potential to compromise network security, by allowing smaller adversarial attacks to be constructed than if the certification had been withheld.
> > >
> > > To elaborate upon this point - it is important to emphasise that certifications do not intrinsically provide security against adversarial attacks. A sample being certified does not tell the user that an attack has not occurred, because *an adversarial perturbation can still be certified*. What a certification does is provide a measure of the distance to the nearest adversarial example, which may be used as part of security assurance systems, but does not provide any direct assurance in and of itself. However, as we demonstrate through this work, releasing information about the certifications gives an attack an information advantage, that can be exploited to construct minimal norm adversarial examples that are smaller - and thus harder to detect - than is possible to construct with equivalent algorithms.
> > >
> > > This is the paradox that we discuss within this paper - that a measure that is associated with increasing security can be also exploited by attackers, to help them minimise the size of their adversarial perturbations, and thus, minimise the risk of detection.
> > >
> > > As to the norm minimising methods, it is true that not all of the tested attacks are formalised as norm-minimising attacks in their original form. However, by constructing our metrics in terms of the *smallest successful adversarial perturbation identified by each technique for a given sample* (as we do), and then tuning our hyper-parameters to minimise the average perturbation norm of the attack across the test set, we do end up being able to minimise the average perturbation norm of the attacks. This is a well established process that has been used in previous attack papers - see for example Carlini & Wagner "Towards Evaluating the Robustness of Neural Networks", which takes this exact same approach.
> > >
> > > We also note that we did use the L2-specific variant of PGD.

---

> > > > ### Comment · Reviewer_Kwhk · 2023-11-23
> > > >
> > > > To emphasize again, the claim that "(releasing) the certificate reduces security" is confusing and misleading. First of all, it is hard to define what "releasing" means here, actually, for the particular evaluation done in this paper, certification is not really "released" but rather achieved using an external algorithm like randomized smoothing given any black-box models.
> > > >
> > > > Although the authors may have their own interpretation, it is better to not make claims like this since multiple reviewers, including me and xpHV, have concerns about the claim. The best way would be considering the constructive feedback provided by reviewers and tuning down the claims, rather than insisting. This is not helpful at all and very disappointing.
> > > >
> > > > The "smaller adversarial example" claim is also questionable. To be clear, the "L2-specific variant of PGD" you mentioned (e.g., projection with L2 norm) is not the specific attack method mentioned (named DDN) in my comment (the paper was not even cited). Please actually read that paper.
> > > >
> > > > I now firmly believe the paper should be rejected.

---

> > > > > ### Author Response · Authors · 2023-11-23
> > > > >
> > > > > As we discuss within the paper, releasing the certification is any form of publishing the certification mechanism that allows the attacker access to this information - either directly by including the certificate with the class prediction, or by producing a class expectation that can be used to infer the certified radius. The idea that certified mechanisms should publish their certifications alongside their predictions has been posited in previous works, motivated by the idea that this would provide an additional measure of confidence to the prediction.
> > > > >
> > > > > What we show is that if the certification is released with the class expectation, then the attacker has an information advantage that can be exploited to construct smaller adversarial examples. This is a clearly stated argument that is made at multiple times through the paper.

---

> ### Author Response · Authors · 2023-11-19
>
> With the review period closing within a few days, we just wanted to take this opportunity to ask if this response had resolved any of your issues with the paper, and to offer our assistance in clarifying any additional points. We'd also note that we have edited the paper (specifically appendix D) to help address the questions regarding the metrics used within the paper.
>
> Thanks in advance

---

### Author Response · Authors · 2023-11-22

With the review period closing shortly, we'd like to thank all the reviewers - we are heartened to see that every reviewer agrees that this paper is novel, interesting, and would make a strong contribution to the literature relating to adversarial risks to machine learning models.

The primary issue that cropped up among the reviewers was a concern that while the chosen evaluations metrics were described in text, the lack of mathematical description lead could lead to potential misunderstandings. In response to this we have submitted a rebuttal revision that adds additional content to the appendices to address these concerns (including adding mathematical descriptions of the evaluation metrics), and have also made changes to the tables and figures to better highlight the levels of out performance that our technique yields relevant to extant attacks.

We hope that by making these changes we have resolved the concerns that reviewers had with the paper, and would hope that the paper can be assessed in line with its technical contributions, as noted and agreed upon by all the reviewers.

If there are any final queries we would be happy to answer them, otherwise thank you all for your time and your consideration.

---

### Meta-Review · Area_Chair_97SJ · 2023-12-14

**Metareview:**

While the proposed attack algorithm itself holds potential interest, the paper's claims and motivations require significant revision. As Reviewers Kwhk and xpHV both pointed out, the proposed attack in this paper focuses on finding adversarial examples slightly outside the certified radius. The existence of such examples doesn't "compromise" the certified defense methods and finding some points outside the radius isn't unexpected. Therefore, while the attack could potentially be reframed as a method for testing certified defense tightness, this requires a major revision of the paper's narrative and claims.

**Justification For Why Not Higher Score:**

The paper may need a major revision on its motivation and claims.

**Justification For Why Not Lower Score:**

N/A

---

### Decision · Program_Chairs · 2024-01-16

Reject